# Borna Disease Virus 1 Phosphoprotein Forms a Tetramer and Interacts with Host Factors Involved in DNA Double-Strand Break Repair and mRNA Processing

**DOI:** 10.3390/v14112358

**Published:** 2022-10-26

**Authors:** Nicolas Tarbouriech, Florian Chenavier, Junna Kawasaki, Kamel Bachiri, Jean-Marie Bourhis, Pierre Legrand, Lily L. Freslon, Estelle M. N. Laurent, Elsa Suberbielle, Rob W. H. Ruigrok, Keizo Tomonaga, Daniel Gonzalez-Dunia, Masayuki Horie, Etienne Coyaud, Thibaut Crépin

**Affiliations:** 1Institut de Biologie Structurale (IBS), CEA, CNRS, Université Grenoble Alpes, 38058 Grenoble, France; 2Laboratory of RNA Viruses, Department of Virus Research, Institute for Life and Medical Sciences, Kyoto University, Kyoto 606-8507, Japan; 3Laboratory of RNA Viruses, Department of Mammalian Regulatory Network, Graduate School of Biostudies, Kyoto University, Kyoto 606-8507, Japan; 4Department of Biology, Univ. Lille, Inserm, CHU Lille, U1192—Protéomique Réponse Inflammatoire Spectrométrie de Masse—PRISM, 59000 Lille, France; 5Synchrotron SOLEIL, L’Orme des Merisiers, 91192 Gif-sur-Yvette, France; 6Institut Toulousain des Maladies Infectieuses et Inflammatoires (Infinity), Université de Toulouse, Inserm, CNRS, UPS, 31024 Toulouse, France; 7Department of Molecular Virology, Graduate School of Medicine, Kyoto University, Kyoto 606-8507, Japan; 8Laboratory of Veterinary Microbiology, Graduate School of Veterinary Science, Osaka Metropolitan University, Izumisano 598-8531, Japan; 9Osaka International Research Center for Infectious Diseases, Osaka Metropolitan University, Izumisano 598-8531, Japan

**Keywords:** *Bornaviridae*, phosphoprotein, structure, interactomics

## Abstract

Determining the structural organisation of viral replication complexes and unravelling the impact of infection on cellular homeostasis represent important challenges in virology. This may prove particularly useful when confronted with viruses that pose a significant threat to human health, that appear unique within their family, or for which knowledge is scarce. Among *Mononegavirales*, bornaviruses (family *Bornaviridae*) stand out due to their compact genomes and their nuclear localisation for replication. The recent recognition of the zoonotic potential of several orthobornaviruses has sparked a surge of interest in improving our knowledge on this viral family. In this work, we provide a complete analysis of the structural organisation of Borna disease virus 1 (BoDV-1) phosphoprotein (P), an important cofactor for polymerase activity. Using X-ray diffusion and diffraction experiments, we revealed that BoDV-1 P adopts a long coiled-coil α-helical structure split into two parts by an original β-strand twist motif, which is highly conserved across the members of whole *Orthobornavirus* genus and may regulate viral replication. In parallel, we used BioID to determine the proximal interactome of P in living cells. We confirmed previously known interactors and identified novel proteins linked to several biological processes such as DNA repair or mRNA metabolism. Altogether, our study provides important structure/function cues, which may improve our understanding of BoDV-1 pathogenesis.

## 1. Introduction

Nonsegmented negative-strand RNA viruses (or *Mononegavirales*) are a class of viruses that infect plants and animals, with many of them causing significant diseases or even death in humans. These enveloped viruses share common strategies in viral replication and/or gene expression [1]. Their genome, encapsulated by the nucleoprotein (N), is the biologically active template for both transcription and replication by the polymerase complex [2] resulting from the association of the viral RNA-dependent RNA polymerase (RdRp or L) with its essential cofactor, phosphoprotein (P). P is a highly versatile protein involved in numerous functions, with a high degree of structural flexibility. Through its modular architecture consisting of long disordered regions that alternate with structured domains, P interacts with the main proteins of the viral RNA synthesis machine. It self-associates in oligomers through a central oligomerisation domain (P_OD_) [3,4,5,6,7,8,9,10,11,12] and bridges the RdRp [13,14,15,16,17,18,19] and the nucleocapsid [20]. It also chaperones N to maintain it in an RNA-free form needed for replication [21,22,23,24,25,26,27,28]. In addition, P is an essential actor of viral replication by recruiting a plethora of cellular partners with proviral activities [10,29,30,31,32,33,34,35,36,37,38,39,40]. The disordered regions of P seem to be of primary importance for these additional functions.

Bornaviruses (family *Bornaviridae*) occupy a special place within *Mononegavirales*. They infect the widest range of species of the animal kingdom, ranging from fishes to mammals [41,42,43]. While most mononegaviruses mainly perform cytoplasmic replication, bornaviruses replicate in the nucleus, where they assemble viral factories [44] designated as “viral Speckles Of Transcripts” (vSPOTs). It has been recently shown that some mammalian orthobornaviruses (*e.g.*, Borna disease virus 1 (BoDV-1) and variegated squirrel bornavirus 1 (VSBV-1)) can infect humans, predominantly affecting neuronal tissues and causing fatal encephalitis [45,46,47]. The genome of bornaviriruses consists of approximately 8900 nucleotides and represents the smallest known genome amongst *Mononegavirales*. BoDV-1 vSPOTs formed by P protein-driven liquid–liquid phase separation [48], are present in the nucleus, in close interaction with the chromatin, where they dock on neuronal DNA double-strand breaks (DSBs), thereby affecting neuronal epigenetics and activity [49]. In particular, both N and P seem to modulate epigenetic signalling in neurons [50,51].

BoDV-1 replicative machinery, a complex between the RdRp (L) and P, produces different mRNAs from the genome via polar sequential transcription, encoding for a total of six viral proteins. Except for the RNA-free N and the matrix protein [52,53], nothing is known concerning the structure of these viral proteins. Here, we used X-ray diffusion and diffraction experiments to provide a complete structural analysis of BoDV-1 P. We observed that P possesses a modular architecture, consisting of a central oligomerisation domain (P_OD_) surrounded by two disordered regions, similar to other *Mononegavirales* Ps. We also showed that BoDV-1 P_OD_ has the most elongated multimerisation domain of all mononegaviruses, with one monomer composed of two consecutive α-helices linked by a short three-amino acid stretch, making a 140 Å long coiled-coil α-helical structure split into two parts by a β-strand twist motif. This particular motif appears to be important for viral replication and is highly conserved across the members of the whole *Orthobornavirus* genus. Moreover, we determined the BoDV-1 P proximal interactome in human cells, confirming previous interactors and identifying novel proteins linked to DNA breaks and mRNA metabolism. Interestingly, the strong overlap between the proximal interactomes in living cells of N- and C-terminal biotin ligase (BirA_R118G_, or BirA*)-tagged BoDV-1 P data in living cell suggests that this structure is highly flexible at the β-strand twist motif and/or that the disordered domains confer enough freedom of movement to the BirA* moiety to biotinylate a highly similar protein environment regardless of its position in the primary protein sequence. Altogether, our results provide important advances in our understanding of the structural organisation of BoDV-1 P protein and strongly support the relevance of our structural model in a living cellular context.

## 2. Materials and Methods

### 2.1. Cloning and Plasmid Preparation

The bacterial expression-optimised DNA coding sequence of Borna disease virus 1 (BoDV-1) P was purchased from GeneArt (Thermofischer Scientific, Regensburg, Germany). Primers were designed for the amplification of the different constructs of BoDV-1 (residues 1 to 201 and 65 to 172), munia bornavirus 1 (MuBV-1; residues 65 to 172), and Gaboon viper virus 1 (GaVV-1; residues 67 to 178). Each DNA fragment was cloned into pETM11 bacterial expression vector (EMBL) using 5′ end *Nco1* and 3′ end *Xho1* restriction sites introduced by PCR amplification. All constructs were expressed as N-terminally TEV-removable His-tagged proteins. Sequencing verifications were performed by Eurofins Genomics.

### 2.2. Protein Expression and Protein Purification

Plasmids with the corresponding constructs were transformed into *Escherichia coli* BL21 RIL (DE3) cells (Life Technologies, Thermo Fischer Scientific, Villebon-sur-Yvette, France). LB cultures were induced for 12 h with 0.3 mM isopropyl-β-D-thiogalactopyranoside (IPTG) at 18 °C, collected by centrifugation, and stored at −80 °C. Briefly, Bacterial pellets were resuspended in 50 mM Tris-HCl (pH 7.5), 500 mM NaCl, 1 mM β-mercaptoethanol (β-ME), and cOmplete^TM^ ethylenediaminetetraacetic acid (EDTA)-free protease inhibitor cocktail (Roche, Meylan, France). After lysis by sonication and centrifugation, the supernatant was complemented with 15 mM imidazole and the recombinant proteins were purified by nickel affinity chromatography (resin Ni-NTA; Qiagen, Courtabœuf, France). Based on SDS-PAGE analysis, fractions of interest were pooled together before the addition of 1% tobacco etch virus (TEV) protease and dialysed overnight in 20 mM HEPES (pH 7.5), 200 mM NaCl, and 1 mM β-ME. A second nickel affinity chromatography was performed to remove both the uncleaved protein of interest and TEV protease before the size-exclusion chromatography (SEC). The flow-through was concentrated by centrifugation (Millipore-Amicon Ultra centrifugal filters 10 K) and injected into a HiLoad^TM^ S200 16/600 column (Cytiva, Velizy-Villacoublay, France) with a NGC system (Bio-Rad, Marnes-La-Coquette, France). All proteins were in the same final buffer composed of 20 mM HEPES (pH 7.5), 150 mM NaCl, and 1mM β-ME.

### 2.3. Crystallisation and X-ray Structure Determination

Crystallisation experiments were initiated using the HTX platform (EMBL, Grenoble, France) before a final manual screening to obtain single crystals. BoDV-1 P_OD_ crystals were obtained from a 10 mg·mL^−1^ protein solution using a precipitant composed of 100 mM HEPES (pH 7.5), 14–18% polyethylene glycol (PEG) 8K, and 200 mM CaCl_2_. MuBV-1 P_OD_ crystals were obtained from a 4 mg·mL^−1^ protein solution using a precipitant composed of 15–18% PEG 3350, 150 mM Mg(NO₃)₂, and 10 mM phenol. GaVV-1 P_OD_ crystals were obtained from a 5 mg·mL^−1^ protein solution using a precipitant composed of 100 mM HEPES (pH 7.5), 18–22% PEG 3350, and 200 mM NaSCN. Crystals were cryo-protected in their mother liquor solution containing 20% glycerol, mounted on cryoloops (Hampton Research, Aliso Viejo, CA, USA), and frozen in liquid nitrogen.

BoDV-1 P_OD_ diffraction experiments were undertaken on ID-29 at the European Synchrotron Radiation Facility (ESRF, Grenoble, France). Data were collected using a Dectris Pilatus 6M detector. Indexing and integration were performed using the XDS program suite [54,55]. Thereafter, data were processed using STARANISO [56], as implemented in autoPROC [57], which applies non-elliptical anisotropic limits based on a locally averaged mean I/σ(I) cut-off, performs a Bayesian estimation of structure amplitudes, and applies an anisotropic correction to the data. Detailed crystallographic statistics are provided in Appendix A. Molecular replacement solutions were obtained with Phaser [58] using the measles virus P_OD_ structure [6]. The initial solution was then used through repeated cycles of autobuilding using Autobuild [59] within the Phenix suite [60]. The final building phase used both manual model building with COOT [61], together with Autobuild cycles and ArpWarp [62,63]. Refinement of the model was carried out using Refmac5 [64].

MuBV-1 P_OD_ diffraction experiments were undertaken on PROXIMA-1 at the SOLEIL Synchrotron (Saint-Aubin, France). Data were collected using a Dectris Eiger-X 16M detector. Indexing and integration were performed using the XDS program suite. Molecular replacement solutions were obtained with Phaser using the BoDV-1 P_OD_ tetrameric structure separated into two ensembles: one containing residues 86 to 125 and the other containing residues 127 to 166. The final building phase used both a manual model building with COOT and refinement of the model using Refmac5.

GaVV-1 POD diffraction experiments were undertaken on PROXIMA-1 for the native dataset and on PROXIMA-2A for the Se-SAD dataset at the SOLEIL Synchrotron. PROXIMA-2A is equipped with a Dectris Eiger-X 9M detector. Indexing and integration were performed using the XDS program suite. Heavy atoms sites were determined using the SHELX suite [65], their positions were refined using PHASER in EP mode, and the phases were improved using PARROT [66] for density modification. The resulting electron density allowed the positioning of an AlphaFold2 [67] model, which was improved and refined against the native dataset with COOT and Refmac5.

The figures were drawn using PyMOL [68].

### 2.4. Small-Angle X-ray Scattering (SAXS) Analysis

All datasets were collected on the SWING beamline at the SOLEIL synchrotron. The collection parameters are listed in Appendix A. The samples were loaded onto a Superdex^TM^ 200 increase 5/150 GL (Cytiva) SEC column and SAXS measurements were performed throughout elution, operating at 20 °C with a flow rate of 0.3 mL·min^−1^ and a mobile phase containing 20 mM HEPES (pH 7.5), 150 mM NaCl, and 5 mM β-ME. The data were integrated and subtracted with Foxtrot [69], providing the first estimation of I(0) and Rg for each frame of the SEC profile. This allowed the identification along the elution peak of the frames with constant Rg values. The corresponding frames were merged into a final scatter curve used for further analysis. Final Rg values were determined using ATSAS [70] and RAW [71]. Dmax values were determined using GNOM [72] in the ATSAS suite and compared to the values obtained with BIFT [73] implemented in RAW. Molecular weights of the proteins were estimated by using the Qr ratio derived from the invariant volume of correlation [74].

### 2.5. Ab Initio Molecular Shape Analysis of P_OD_

*Ab initio* shape analyses were performed using DAMMIF [75] from the ATSAS package. Twenty independent DAMMIF calculations were performed both with and without four-fold (P4) symmetry. In each case, to create the most probable molecular shape, an average shape was determined with DAMAVER [76], with and without P4 symmetry. The filtered models were then reprocessed against the experimental data using DAMMIN [77].

### 2.6. Full Atom Modelling of BoDV-1 P_FULL_

An initial pool of 10,000 random conformation models of BoDV P_FULL_ was generated using the Ensemble Optimisation Method (EOM 2.1 [78]). The models generated for the disordered N-terminal (residues 1 to 71) and C-terminal segments (residues 167 to 201) were attached to the P_OD_ X-ray structure. As the disordered parts generated by EOM are Cα, we used an in-house procedure to obtain the full atom structures of the pool. First, PD2 ca2main [79] was used to reconstruct the backbone, followed by SCWRL4 [80] to add the side chains. This pool was then used to select a final ensemble with the EOM algorithm based on the experimental scattering curve. This process was repeated three times.

### 2.7. Minireplicon Assays

Minireplicon assays were conducted as described previously [81]. Briefly, 293T cells seeded onto 48-well plates were transfected with 50 ng of minigenome plasmid containing Gaussia luciferase as a reporter gene [82], 50 ng of pCAGGS-N, 5 ng of pCAGGS-P (or point-mutated P), and 50 ng of pCAGGS-L using TransIT293 (Mirus) according to the manufacturer’s instructions. The culture supernatants were collected at 72 h post transfection and subjected to luciferase assay using a Biolux Gaussia luciferase assay kit (New England BioLabs, Evry, France) according to the manufacturer’s instructions. The assays were performed in technical duplicates for each experiment.

### 2.8. Determination of Proximal Host–Viral Protein Interactions in Living Cells (BioID)

These experiments were performed as in [83]. Briefly, Flp-In™ T-REx™ HEK293 cells (Life Technologies) cells were co-transfected with pOG44 (encoding the Flp recombinase) and pcDNA5 FRT/TO FlagBirA*- or BirA*Flag P_FULL_ (N- and C-ter tag, respectively) of BoDV-1. Cells were selected with Hygromycin B to generate new stable tetracycline-inducible cell lines expressing single copies of N- and C-terminal tagged P_FULL_-BirA* fusion proteins. Cultures were expanded to ~100 million cells per sample, before being placed into a medium of 1 µg·mL^−1^ tetracycline to induce bait protein expression, along with 50 µM biotin. After 24 h, induction cells were collected, washed twice with PBS, spun down, and stored at −80 °C. Each pellet was resuspended in a 5 mL of modified RIPA buffer (50 mM Tris-HCl (pH 7.5), 150 mM NaCl, 1 mM EDTA, 1 mM EGTA, 1% Triton X-100, 0.1% SDS, 1:500 protease inhibitor cocktail (Sigma-Aldrich, Saint-Quentin Fallavier, France), and 1:1000 Turbonuclease (BPS Bioscience, Nanterre, France)), incubated for one hour on an end-over-end rotator at 4 °C, sonicated until no visible aggregates remained, and centrifuged at 45,000× *g* for 20 min. Cleared supernatants containing all compartment solubilized proteins were transferred to new tubes and incubated with 30 µL of packed and pre-equilibrated streptavidin resin (Ultralink, Pierce, Thermo Fischer Scientific, Villebon-sur-Yvette, France; 2 h under rotation at 4 °C) to capture biotinylated species. Beads were pelleted (300 g, 1 min) and washed six times with 50 mM ammonium bicarbonate (AB). Trypsin (1 µg) was added to each 200 µL sample to digest on beads (overnight at 37 °C). Tryptic peptides were collected in the supernatant, lyophilized, and desalted on C18 ziptips. One-quarter of each sample was analysed by nLC MS-MS (Easy1000 in line with a Thermo Q-Exactive; see Appendix A). The MS raw files were then analysed using MaxQuant (version 1.5.8.3) against the UniProt Reference database to generate the protein identification list with a 1% FDR. High confidence proximal interactors were defined using Perseus (version 1.6.15.0) with the following filters for each given ID: peptides detected in 3/3 bait protein runs; an average Log_2_ LFQ abundance against average top 3/20 control over 1; and a corresponding q-value < 0.01 (Appendix A).

## 3. Results

### 3.1. X-ray Structure of the Oligomerisation Domain of BoDV-1 Phosphoprotein

With *Mononegavirales* Ps being known as multimodular nonglobular molecules, we first performed a bioinformatic analysis of the BoDV-1 P amino acid sequence using 16 web servers, all of them predicting the location of disordered regions and secondary structure elements. We then combined all of the results in a single representation (D-score; Figure 1a). On this graph, residues with a D-score <0.50 were assigned as disordered, while residues with a D-score >0.5 were assigned as structured. We observed that BoDV-1 P presented an organisation similar to other *Mononegavirales* Ps, with central folded domains flanked by N- and C-terminal disordered regions. Structure prediction using AlphaFold2 is compatible with the D-score analysis (Appendix A). Based on this prediction, two constructs were designed (Figure 1b): the full-length P (P_FULL_) and the central domain (from residue 65 to 172) that we hypothesized to be the oligomerisation domain (P_OD_), even if its size represented nearly half of the protein length and was much longer than P_OD_ from other viruses.

BoDV-1 P_OD_ crystals (fine plates growing in urchins) were obtained in PEG 8000, only in the presence of calcium chloride. Crystals belonged to space group *P42_1_2*, with cell parameters a = b = 35 Å and c = 166 Å. The analysis of the asymmetric unit indicated that it could only contain a single-amino acid chain (Matthew coefficient of 2.1 Å^3^·Da^−1^ with a solvent content of 42% for a 108 amino acid-long chain), meaning that it was likely to contain a single narrow elongated molecule. This informed our choice to use a single coiled-coil chain of the measles virus tetrameric P_OD_ [6] as a starting model to solve the structure by molecular replacement. An initial solution was obtained from Phaser [58] and improved using Autobuild [59], before being classically built and refined. The final structure is shown in Figure 2a.

The monomeric P_OD_ is composed of two consecutive aligned α-helices (from residue E73 to Q123 and S128 to A164, respectively) linked by a short stretch of three residues, making a ≈ 140 Å long molecule. As most *Mononegavirales* P assemble in oligomers through their central domain, we analysed P_OD_ by SAXS (Appendix A) in order to compare its behaviour in the crystal and in solution. With a Dmax of 170 ± 9 Å, it confirmed one dimension of the protein. Furthermore, the volume of correlation (Vc) determination method [74] estimated a Mw of 40 kDa, corresponding to a tetramer (discussed later in Section 3.3). A tetrameric P_OD_ could then be reconstituted by applying the inherent 4-fold symmetry of this tetragonal space group (Figure 2b). It was then composed of two α-helical coiled-coils with all the monomers being parallel, with a break centred at position 126 in the middle of the whole structure (Figure 2d and Appendix A). Considering the electrostatic surface of the domain, the break seemed to separate the N-terminal acidic region from the C-terminal more neutral region (Figure 2c). A careful analysis of the Crick parameters [85,86] using CCCP [87] and TWISTER [88] demonstrated a decrease in the helical and superhelical radius together with an increase in the rise per residue and a decrease in the periodicity. In addition, the Ramachandran plot showed that residue D126 fell into the β-area, whereas neighbouring residues (C125 and H127) were in intermediate positions, between the α- and the β-areas (Appendix A). Altogether, our data point towards the formation of a β-strand twist motif within the helices, as described for trimeric oligomerisation domains [89]. As opposed to the trimers, the tetrameric motif of P_OD_ identified herein did not form regular β-sheet main chain hydrogen bonds, and the shift in the helical axis was limited to 45° (Figure 2e) instead of the 120° observed in trimers.

### 3.2. The β-Strand Twist of BoDV-1 P Is Important for Viral Replication

In order to assess the involvement of the β-strand twist motif in BoDV-1 replication, we performed minireplicon assays using a series of P mutants, each one bearing a single amino acid substitution at the twist or in its proximal regions (from position 124 to 129). 293T cells were then transfected with a minigenome plasmid containing the Gaussia luciferase as a reporter gene, together with plasmids encoding for N, L, and different versions of P. The Gaussia luciferase activity levels were then measured (Figure 3a). Correct expression of each P mutant was confirmed by western blotting (Figure 3b). Interestingly, substitutions by alanine of R124 and D129, two proximal residues of the motif, did not affect the polymerase activity. Rather, the activity was abolished or markedly reduced by mutating C125, D126, or H127. Note that the quantity of protein for this mutant, as well as for the second mutant tested for this position (*i.e.*, C125D), appeared strongly reduced in the western blot analysis. Altogether, these results suggest that the β-strand twist motif is important for BoDV-1 replication, notably concerning D126 and H127 residues, whereas C125 would have an effect on the stability and/or the folding of P. Cysteine residues are known to form disulfide bridges and stabilise the structuration of secreted proteins. Surprisingly for a nuclear protein, such a disulfide bridge has been previously proposed at position 125 of BoDV-1 P [90]. Our experimental conditions did not allow us to detect them, especially in the crystal structure. However, compared to D126, which is strictly conserved (Figure 4j and Appendix A), the amino acid in position 125 is a cysteine only in mammalian 1 orthobornaviruses (*i.e.*, BoDV-1 and -2).

### 3.3. The β-Strand Twist Motif Is Conserved in Orthobornavirus P_OD_

The comparison with tetrameric oligomerisation domains of P from other *Mononegavirales* (metapneumovirus, parainfluenza 5, measles, mumps, Nipah and murine respirovirus) showed that BoDV-1 P_OD_ was indeed the longest (the size of one monomer in the corresponding X-ray structures is 141, 100, and 93 Å long for BoDV-1, murine respirovirus, and measles P_OD_, respectively). Above all, it is the only one with such a β-strand twist-like motif that breaks the continuity of its α-helical coiled-coil organisation. We therefore investigated its conservation in other members of the *Bornaviridae* family. Several P_OD_ were cloned, expressed, and purified. Crystals were obtained for the P_OD_ of munia bornavirus 1 (MuBV-1; 77% sequence identity with BoDV-1) and Gaboon viper virus 1 (GaVV-1; 41 and 43% sequence identity with BoDV-1 and MuBV-1, respectively). The complete SEC-SAXS and X-ray analysis of the three P_OD_ were performed (Figure 4).

In solution, the three P_OD_ presented the same behaviour. They were all eluted as a single straight symmetric peak from the SEC-SAXS column, without any aggregate signal. Using the Vc determination on the diffusion data, the MW were estimated between 40 and 45 kDa, 12–16% lower than the theoretical values (Appendix A), all corresponding to tetramers. The Guinier transforms measured a hydrodynamic radius (Rg) between 44 and 46 Å and GNOM produced the pair distribution functions that fit with Dmax between 169 and 178 Å. Independently to the X-ray crystallography analysis, *ab initio* modelling was performed using twenty DAMMIF models averaged by DAMAVER. Averaged models’ correlation with the diffusion curves were then checked using the damstart file as a starting envelope for DAMMIN (Figure 4a–c and Appendix A). Each DAMAVER envelope appeared as an elongated tube (Figure 4d–f). In parallel, the X-ray structures of MuBV-1 and GaVV-1 were solved using different determination procedures (Appendix A). The structure of MuBV-1 P_OD_ was solved by molecular replacement using BoDV-1 P_OD_ as a starting model, whereas GaVV-1 P_OD_ was obtained de novo using Se-Met derivative crystals. For both MuBV-1 and GaVV-1, the asymmetric units contained a tetrameric P_OD_, whereas the tetramer for BoDV-1 could be obtained after applying the inherent symmetry operations of the tetragonal space group. Figure 4g–i shows the three biological P_OD_ (*i.e.*, as tetramers) in the same orientation. The three X-ray structures fit perfectly with the SAXS data. MuBV-1 and GaVV-1 P_OD_ presented similar architectures composed of two α-helical coiled-coils (the monomers being parallel) linked by a kink. Globally, the only differences concern the relative spatial position of the two coiled-coils, due to the inherent flexibility induced by such a break (Appendix A) and the crystal packing. MuBV-1 and GaVV-1 breaks were also composed of three residues, with a strictly conserved aspartate in the middle (Figure 4j and Appendix A). The Ramachandran analysis showed that all of them fall into the β-area, with neighbouring residues at intermediate positions (Appendix A); this confirms that this specificity is conserved in all Orthobornavirus P.

### 3.4. SAXS Analysis of Full-Length BoDV-1 Phosphoprotein

Having shown that P_OD_ assembles as a tetramer, we next analysed BoDV-1 P_FULL_ by SAXS in order to provide a complete view of the protein (Appendix A). A P_FULL_ diffusion curve is shown in Figure 5a. Data processing (Figure 5b and Appendix A) indicated that it is elongated for a 201 amino acid-long protein, with a Dmax value of 215 ± 10 Å, and Vc determination gives a value of 90 kDa, in accordance with a tetramer in solution, similar to the behaviour of P_OD_. The main difference between P_OD_ and P_FULL_ became apparent when examining the Kratky plots. Indeed, while a nice bell-shaped (Gaussian) peak was observed for P_OD_ (Appendix A), we obtained a curve with a plateau that slowly decays for P_FULL_ (Figure 5c). These two profiles are characteristic of compact/globular and partially unfolded proteins, respectively, thereby validating our bioinformatic analysis. These data also indicate that the biological unit of BoDV-1 P forms a tetramer in solution, with a central domain responsible for this oligomerisation.

Being partially unfolded, the inherent flexibility of BoDV-1 P complicated the analysis of its structural organisation as a single particle. In order to progress in the complete structural analysis of P_FULL_, we used a modelling procedure based on EOM [78] and the P_OD_ X-ray structure discussed in Section 3.2. A total of 10,000 Cα chains corresponding to the missing N- and C-terminal parts were selected to satisfy the SAXS data. Each protein backbone chain was reconstructed from these Cα models [79], and the missing side-chains were finally added with SCWRL4 [80]. Figure 5d shows an ensemble of 6 (out of the 10,000) independent models that fit the SAXS diffusion curves (black line in Figure 5a). This model revealed that despite its rigid central part, BoDV-1 P displays significant flexibility, as illustrated in the figure by the superimposition of the different conformers. However, transitory and/or stable small secondary-structure elements within both N- and C-terminal extremities cannot be excluded in our modelling.

### 3.5. Determination of BoDV-1 P–Host Proximal Interactomics in Living Cells

To gather insights in P_FULL_ behaviour in human living cells, we identified its proximity interactors using the BioID technique [91]. BioID uniquely provides information on physical and proximal protein-of-interest partners through the conjugation of biotin on the vicinal free ε-amine groups of surrounding proteins (within a 10 nm radius [92]). This covalent modification renders possible the thorough solubilisation of all cell compartments using harsh lysis conditions since protein interactions no longer need to be preserved during the purification process. Briefly, we performed BioID using stable and tetracycline-inducible N- and C-terminal BirA* tagged P_FULL_ HEK293 TREx Flp-In cells and purified the biotinylated proteins on streptavidin columns.

We then identified the proximal interactomes by nLC-MS/MS. Following background removal using a set of FlagBirA* alone samples (localising both in the cytoplasm and the nucleus [93]), we identified a total of 207 high-confidence interactors (Appendix A; 147 with FlagBirA*-P_FULL_ and 154 with P_FULL_-BirA*Flag). As expected, we identified previously reported P_FULL_ interactions (*e.g.*, HMGB1 [44] and DNA break repair machineries), but the vast majority of our dataset was novel and links P_FULL_ to new mechanisms/complexes. In the set of more than 207 interactors (either nuclear or cytoplasmic), 123 were strictly nuclear, 55 could be in both compartments, and 6 were cytoplasmic (Appendix A). These findings are in agreement with the intrinsic ability of P_FULL_ to localise in the nucleus. The relevance of the non-nuclear interactors identified here is subject to caution and could be model-dependent, as discussed hereafter. Word clouds of Gene Ontology enrichment analyses indicated that P could associate with diverse nuclear subcompartments (Figure 6a; *e.g.*, telomeres, nuclear speck, kinetochores) and with factors involved in transcription and DNA repair processes (Figure 6b). Further exploring our data, P_FULL_ BioID identified 44 chromatin-associated partners, with 10 and 22 of them being linked to either replication forks or double-strand break repair, respectively (Appendix A; Figure 6c). P_FULL_ BioID also detected 52 partners linked to RNA metabolism, including 13 components of the U2-type spliceosomal complex and 6 members involved in mRNA polyadenylation. A comparison between N- and C-terminal BirA* tagged P_FULL_ (Appendix A; Figure 6d) revealed that N-terminal BirA* preferentially identified the HMGB1 and HMGB2 proteins, with HMGB3 being just below the threshold (*p* = 0.014) [94,95], as well as dynactin proteins (DCTN2 and DCTN4). This observation suggests two possibilities: (i) HMGB1/2 and DCTN2/4 proteins interact at the N-terminal region of the BoDV-1 phosphoprotein, with a putative structural “locking” of the twist or of the N-terminal disordered region; (ii) C-terminal tagging of P could hinder these interactions. Conversely, C-terminal tagged P_FULL_ preferentially identified two components of the Commander complex (COMMD3 and CCDC22), which are involved in the regulation of NFκB signalling regulation [96]. Similarly, it could indicate an association of these factors with the C-terminal region of P_FULL_, potentially concomitant with an induced structural stiffness, or that the N-terminal tagging impairs their binding or proximity to P_FULL_. Finally, several additional proximal partners were associated with the regulation of innate immune responses (*i.e.*, PQBP1 [97] and its paralog CACTIN [98], HEXIM1 [99], and METTL3 [100]), suggesting a putative function of P in BoDV-1-driven innate immune escape.

## 4. Discussion

By interacting with all the proteins of the replication machinery and recruiting many cellular partners, *Mononegavirales* P can be considered as the cornerstone of viral proliferation. Their peculiar architecture consisting of long disordered regions that alternate with structured domains is largely related to all these functions. Our structural findings reveal that orthobornaviral P does not deviate from that model, with an extended central oligomerisation domain that represents half of the size of the protein, surrounded by N- and C-terminal flexible parts. In the past, it was speculated that BoDV-1 P assembled in oligomers, from dimers stabilized by an internal disulfide bridge to tetramers [90,101]. With our data, we confirm that orthobornaviral P forms tetramers in solution, which is in complete agreement with these previous works. There were uncertainties regarding the formation of a trimeric or a tetrameric form in these previous studies, which could be related to the fact that the qualitative analysis of such a protein with disordered propensities was performed using a Superose 12 column [101]. Indeed, disordered and globular proteins are known to be eluted differently from SEC, and the column used at that time was less suited to efficiently discriminate proteins with such small differences in composition. Because of our experimental procedures, in particular during the crystallisation step, we cannot totally exclude the presence of a disulfide bridge at position 125 for a non-secreted protein [90]. However, the presence of a disulfide bridge at position 125 may not be of primary importance, since C125 is only conserved in mammalian 1 orthobornaviruses P (Appendix A). For the other orthobornaviruses, position 125 (or similar) is occupied by amino acids with different physico-chemical properties (from serine/threonine to leucine/isoleucine). In contrast, D126 is strictly conserved amongst the members of genus *Orthobornavirus*, and our minireplicon assays suggested that it plays an important role in viral replication, without any explanation yet. Future studies examining the role of D126 position in the assembly of the polymerase complex and vSPOT formation will help to better understand the role of this key position.

In *Mononegavirales*, *Bornaviridae* P is the shortest regarding amino acid composition. However, it bears the most elongated oligomerisation domain, although to date, this feature has only been proven in solution, far from the cellular context. Structural data on the *Bornaviridae* L-P complex may be particularly relevant for information about P structure within the replication complex. For instance, P may conserve its elongated structuration, similarly to PIV5, where the tetrameric P_OD_ protrudes from the globular RdRp [15]. Alternatively, the twist present in BoDV-1 P_OD_ may correspond to a hinge that would favour a structural rearrangement of the cofactor to reach an octopus-like organisation intertwined with L, as seen for RSV [14,102]. However, such a rearrangement would involve the destabilisation of the N- or the C-terminal coiled-coil, which would be energetically consuming. The L binding site for P has been described to be located between residues 135 and 183 [101]. This part of the P_OD_ corresponds to the most conserved of the two helical coiled-coils (the sequence conservations of the N- and C-terminal α-helixes are 30% and 55%, respectively). The C-terminal α-helix ending at position 162 in our crystal structure, the interaction between L and P could be only mediated by the end of the second α-helix and a stretch of flexible residues up to 183, without any major rearrangement of the cofactor.

Our results also show that BoDV-1 P presents significant inherent flexibility through its N- and C-terminal parts. This is stressed by the high overlap between the N- and C-terminal BioID tagged P proximal interactomes. Comparing the N- and C-terminal BioID tagged proximal interactomes, we observed that only 45% (94/207) of factors are in common. However, when comparing the relative abundance of the high-confidence interactors, over 90% (189/207) did not show significant differences depending on the tag positioning. This apparent discrepancy is due to stringent filtering used to define high-confidence proximal interactors, and the LFQ comparison between merged hits strongly supports the ability of the BirA* moiety to conjugate biotin on proteins located in similar places, whether it is positioned at the N- or C-terminal of P. Given that the biotinylation radius of BirA* has been estimated to be ~10 nm [92], and that that of P_OD_ alone is ~14 nm, we hypothesise that the disordered domains remain unstructured in living cells and confer to BirA* enough freedom to move around the P_OD_, thereby biotinylating similar species. Our analysis of the proximal interactomes confirmed the intrinsic ability of BoDV-1 P to localise at DNA DSBs regions. The interactome analysis particularly pointed out a proximity with sensing factors (TP53BP1) or resolution by repair of the DSB (*i.e.*, NBN, BRIP, PPP4). It also supported its role in recruiting numerous mRNA processing factors, which could serve as complementation factors for vRNA production and maturation in the context of BoDV-1 infection (*e.g.*, mRNA polyadenylation machinery). Considering that BioID identifies proximal partners of the fusion bait protein over 24 h (labelling time roughly equal to one cell cycle in HEK293), our data hence represent a collection of all detectable proteins that were at some point in contact with or in the close vicinity of P_FULL_. Considering that differentiated neurons (the cells targeted by BoDV-1) are postmitotic cells and thus do not enter in mitosis, it is important to stress that our approach must be seen as a discovery step with several limitations to keep in mind. Besides lacking the co-expression of N and L BoDV-1 proteins, which could trigger synergic interactions, labelling over a cell cycle time could explain the presence of a few non-nuclear factors that could interact with BoDV-1 P upon nuclear envelope breakdown (prophase). If this hypothesis is true, non-nuclear interactors identified here should be considered with caution since P expressed alone would remain strictly nuclear. However, since P is translated in the cytoplasm, these interactions could represent P processing factors before its delivery to the nucleoplasm. In line with the previous remark, we also identified replication fork factors. In view of the tropism of BoDV-1 towards neurons, our data regarding the replication fork components must be considered with caution as well. Given the recruitment of DNA repair machineries on stalled forks during replication stress to repair DNA damage and resume replication ([103] for review), we consider the interactions with the replication fork factor to be model-dependent and not necessarily relevant in an infectious context. However, the frequent occurrence of DNA DSBs and the peculiar centrality of the DNA repair machineries in driving neurons somatic mosaicism and diverse neurodegenerative diseases are well documented (reviewed in [104]). We see our data as a discovery step sketching the ability of BoDV-1 phosphoprotein to interact with ubiquitous DNA repair factors. Moreover, these results may be particularly important in the context of novel findings showing that neuronal DSB responses play key roles in learning and memory [105,106]. For instance, our data could serve as a first step to supervised studies in infected neurons aiming at understanding which of these factors are essential for the positioning of vSPOTs at DNA DSBs. In addition, the identification of proximal partners involved in the regulation of innate immune responses suggests that P could interfere with the host cell’s ability to defend against BoDV-1 infection.

Beyond supporting the relevance of our structural model in living cells, our proximal interactomics data represent—to our knowledge—the first published BioID study of any *Mononegavirales* P. Making sense of what is suspected in the field and providing dozens of nuclear ubiquitous factors, this work hence represents an unprecedented wealth of data, which could direct numerous studies aiming at deciphering BoDV-1 P functions in human cells. Repeating such experiments in an infectious context (*i.e.*, complementing with the other BoDV-1 proteins ideally in neurons) would provide information on the cooperation between viral proteins and better resolve the issue of vSPOT composition.

## Figures and Tables

**Figure 1 viruses-14-02358-f001:**
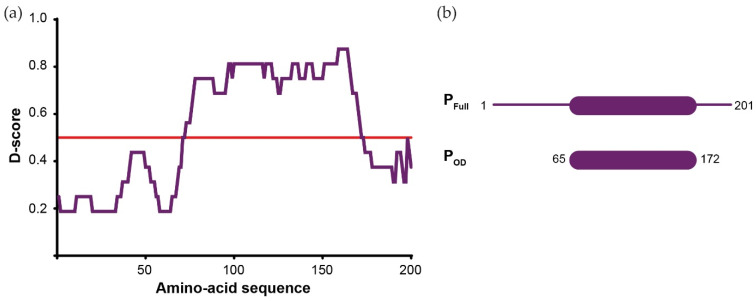
Predicted architecture of BoDV-1 phosphoprotein. (**a**) D-score (score for disorder as a function of residue) of BoDV-1 P. The prediction was based on 16 predictor web servers and the D-score was calculated by adding the values for each residue and dividing by the number of used algorithms. We arbitrarily defined a threshold level at 0.50. Residues with a D-score <0.50 were assigned as disordered [84]. (**b**) Scheme of the BoDV-1 P constructs designed for this work based on the bioinformatics analysis.

**Figure 2 viruses-14-02358-f002:**
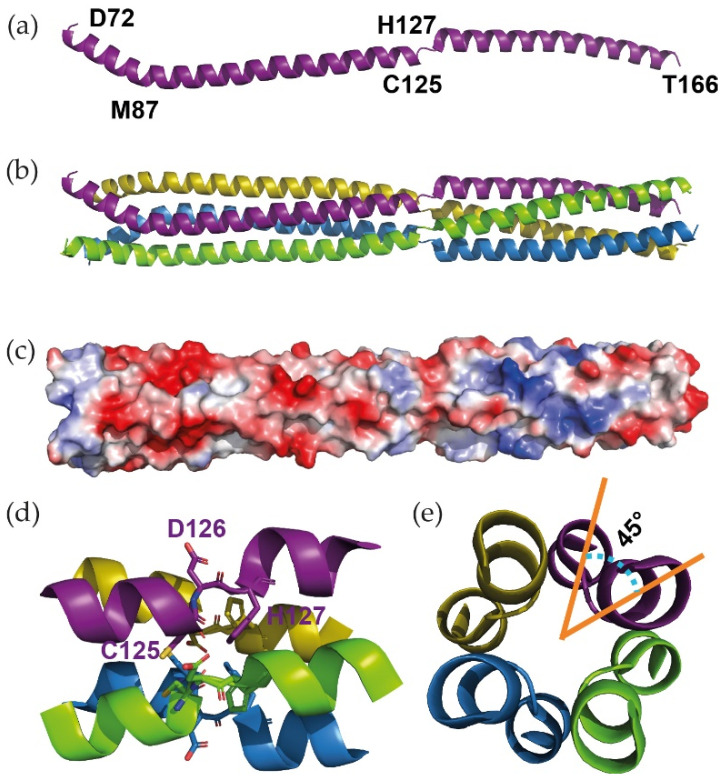
X-ray structure of BoDV-1 P_OD_. (**a**) Structure of BoDV-1 monomeric P_OD_ as obtained in the asymmetric unit. (**b**) Structure of the tetrameric P_OD_ after applying the inherent symmetry operations of the tetragonal space group. The four parallel monomers are coloured in purple, yellow, blue, and green, respectively. (**c**) Electrostatic surface potential of P_OD_. The potential scale ranges from −5 kT/e (red) to 5 kT/e (blue); the orientation is as in (**b**). (**d**) Detail of the β-strand twist motif that separates the two α-helical coiled-coils. (**e**) Orthogonal view of the β-strand twist motif as shown in (**d**), showing the 45° rotation of the helical axis around the super helical axis.

**Figure 3 viruses-14-02358-f003:**
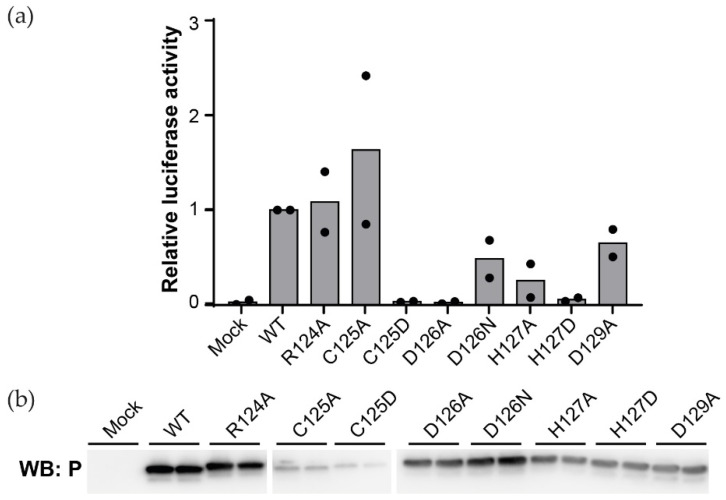
Mutations within the β-strand twist motif of BoDV-1 P modulates viral replication. (**a**) Minireplicon assay using a series of P mutants. The relative luciferase activities are shown as the ratios relative to wild-type P transfected cells (WT). The indicated BoDV-1 P mutants were used as helper plasmids. The dots show mean values (biological replicates; n = 2). (**b**) Detection of BoDV-1 P mutants by western blotting. The whole western blot is shown in Appendix A.

**Figure 4 viruses-14-02358-f004:**
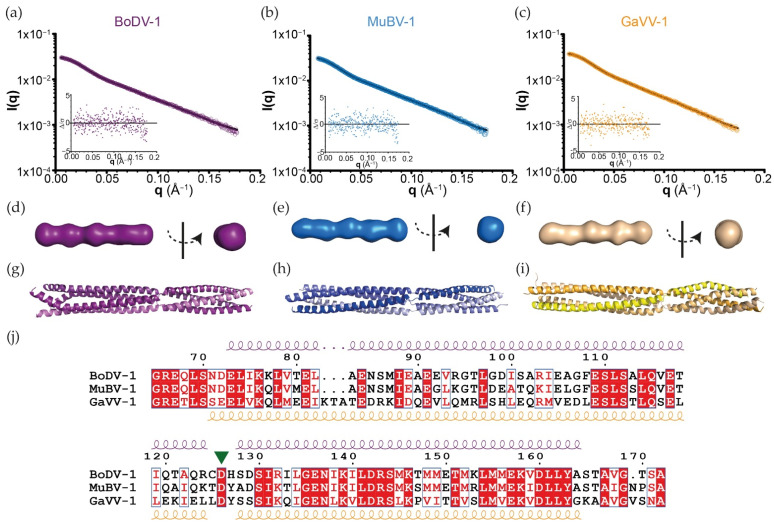
Structural analysis of orthobornaviral P_OD_. Experimental SAXS diffusion curves (with residuals (small plots)) of (**a**) BoDV-1, (**b**) MuBV-1, and (**c**) GaVV-1 P_OD_ overlaid with the averaged SAXS curves (black) resulting from the *ab initio* models shown in panels **d**–**f**. Low-resolution envelops of (**d**) BoDV-1, (**e**) MuBV-1, and (**f**) GaVV-1 P_OD_ based on the *ab initio* modelling from the SAXS data. X-ray structure of (**g**) BoDV-1, (**h**) MuBV-1, and (**i**) GaVV-1 P_OD_ with the same orientation (*i.e.*, N-terminal on the left). The SAXS envelops (**d**–**f**) and the X-ray structures (**g**–**i**) are not at the same scale. Appendix A presents the dockings of the X-ray structures in the SAXS envelops. (**j**) Sequence alignment of the corresponding P_OD_ with the secondary structures of BoDV-1 and GaVV-1, shown above and below the sequence alignment, respectively. Residues in a solid red background are identical between the three sequences. The green triangle indicates the position of the conserved aspartate residue within the β-strand twist motif.

**Figure 5 viruses-14-02358-f005:**
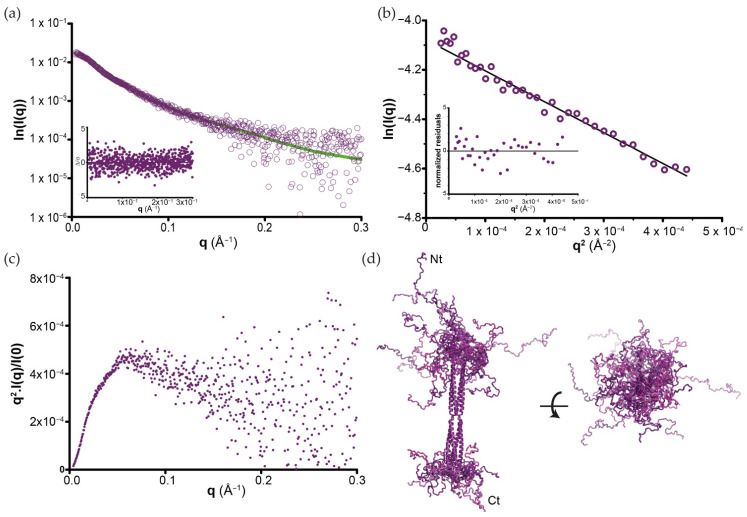
Flexibility analysis of BoDV-1 full-length P. (**a**) Experimental SAXS diffusion curve (deep-purple circles) of P_FULL_ overlaid with the averaged SAXS curves (green) calculated from the EOM ensemble shown in panel (**d**). (**b**) Guinier extrapolation for the determination of the Rg. The smallest plot corresponds to the normalised residual. (**c**) Kratky plot showing the partially unfolded propensity of P_FULL_ samples. (**d**) Conformational ensemble (6 of 10,000 conformers are shown; these models are perfectly representative of the whole population (Appendix A)) of P_FULL_ obtained from the modelling procedure based on EOM. P_OD_ X-ray structure is coloured deep purple, whereas the N- and C-terminal intrinsically disordered regions are coloured with six different shades of purples.

**Figure 6 viruses-14-02358-f006:**
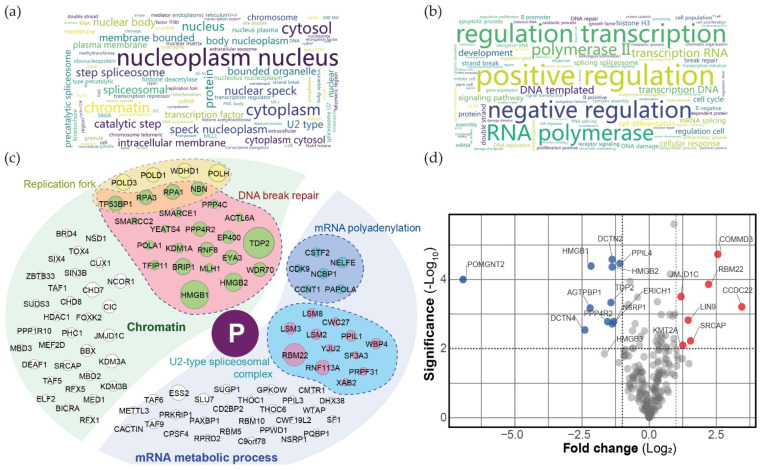
Proximal interactomes of P_FULL_ in human cells. Word clouds of Gene Ontology enrichment analysis (https://toppgene.cchmc.org/; accessed on 1 September 2022) of Cellular Component (**a**) and Biological Process (**b**) (assembled on http://www.bioinformatics.com.cn/plot_basic_wordcloud_118_en; accessed on 1 September 2022). (**c**) Illustration of a subset of P_FULL_ high-confidence interactors. Proteins are grouped and coloured by selected functions or compartments, as indicated on the diagram; node diameter is proportional to the Log_2_ fold change against control. (**d**) Volcano plot comparing the N- and C-terminal BirA* tagged BioID data. Blue and red dots represent N- and C-terminal BirA* tagged preferential partners, respectively (*p*-value < 0.01 and Log_2_ fold change > 1 between both fusions). Assembled using https://huygens.science.uva.nl/VolcaNoseR/; accessed on 1 September 2022). All data are based on three biological replicates and have been filtered for background against a set of 20 control runs (see Appendix A for details).

## Data Availability

The X-ray diffraction data are available in the protein databank (https://www.rcsb.org/; accessed on 1 September 2022) with the accession numbers 8B8A, 8B8B, and 8B8D for BoDV-1, MuBV-1, and GaVV-1 P_OD_, respectively; the SAXS data can be found in the small-angle scattering biological databank (https://www.sasbdb.org/; accessed on 1 September 2022) with the accession numbers SASDQP5, SASDQQ5, SASDQR5, and SASDQS5 for BoDV-1 P_FULL_, BoDV-1, MuBV-1, and GaVV-1 P_OD_, respectively; the mass spectrometry proteomics data have been deposited to the ProteomeXchange Consortium via the PRIDE [107] partner repository with the dataset identifier PDX036750 at http://www.ebi.ac.uk/pride; accessed on 1 September 2022 (login: reviewer_pxd036750@ebi.ac.uk; pw: OXUWT9se).

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
