# Peer review of "Borna Disease Virus 1 Phosphoprotein Forms a Tetramer and Interacts with Host Factors Involved in DNA Double-Strand Break Repair and mRNA Processing"

_viruses, 2022, doi:10.3390/v14112358_

Round 1

Reviewer 1 Report

This is an interesting study providing the first structural data on Bornavirus P proteins, integrating crystallographic and small angle X-ray scattering measurements. The authors demonstrate that the oligomerisation domain )POD) of three different bornaviral P proteins adopt the same tetrameric coiled-coil arrangement, with a conserved break around residue Asp126 (BoDV-1 residue numbering used here throughout) that results in a 45 degree twist (perhaps a better description than "kink"?) about the coil axis. In the full length protein, the central POD coiled coil is flanked by intrinsically disordered N- and C-terminal peptides.

This is a nice piece of work that is certainly worthy of publication. There are some issues that in my opinion the authors should address:

I find the nomenclature beta-layer-like a little misleading, as this actually only pertains to the phi/psi angles of Asp126; no beta-structure is found (in contrast to that described for trimeric alpha/beta coiled coils). According to Figures 4 and S8, residue 124 is also no longer part of the helix; why is this not included as part of the twist? Do the authors have any explanation for the presence of this twist? It seems to be associated with conserved Asp126, but I do not see a clear structural reason why this should be so. If it is also not clear to the authors, they should state this explicitly. Do any of the mutants analysed show changes in stability to chemical or thrmal denaturation?

It would be nice to see the electron density in the neighborhood of the break in the three structures, which could be in the supplementary data. Were each of the four chains in the asymmetric units of the MuBV-1 and GaVV-1 POD crystals absolutely identical, or were there deviations from fourfold symmetry? It is stated several times that the twist/kink results in "flexibility". Apart from the BioID data (see below for further comments on this), I do not see any evidence for structural flexibility - perhaps a superposition of the three tetrameric arrangements shows this?

How is the three residue insert after BoDV Pro83 accommodated in GaVV-1? Is the coiled coil simply extended?

The minireplicon data suggest that residues in the twist/kink region play a role in viral replication. This is an interesting aspect, but would be more convincing if biological (in addition to technical) replicates were presented.

The only part of the manuscript that I find less than convincing is the BioID data. The authors themselves agree that their results may lack validity due to use of HEK293 (rather than neuronal) cells and the lack of viral N and L proteins. I would be inclined to remove these data from the paper.

Reviewer 2 Report

Tarbouriech et al. show X-ray crystal structures of the oligomerization domain of the  polymerase cofactor (P) proteins from three bornaviruses: BoDV-1, MuBV-1 and GaVV-1. Supported by small-angle X-ray scattering in solution, the crystal structures reveal that P forms a tetramer. The oligomerization domain is larger than in related viruses, and the continuity of its alpha-helix is broken at a position near its center. Single point mutations within this break region are shown to impact transcription/replication efficiency in a minireplicon system. In addition to this structural investigation, proximity labeling in HEK293 cells reveals previously undiscovered potential interaction partners of the BoDV-1 P protein.

I enjoyed reading this manuscript and believe it will be of interest to many virologists, in particular those interested in the replication and transcription of negative-stranded RNA viruses. I have a few minor comments:

1) line 241: "Structure prediction using AlphaFold2 is compatible with the D-score analysis."

I would be curious to see this AlphaFold prediction in a supplementary figure, ideally a prediction for the tetrameric assembly. How similar is it to the experimental structure? Does it also show the "beta-layer"?

2) line 252: "BoDV-1 POD crystals (fine plates growing in urchins) were obtained in PEG, only in the presence of magnesium."

This wording seems unnecessarily imprecise. I suggest to either describe the exact crystallization condition (which PEG? which magnesium salt? what were the concentrations?) or to simply refer to the methods section without giving any incomplete description here.

3) line 301: "Surprisingly, the C125A mutant displayed an increased activity, although the quantity of protein for this mutant, as well as for the second mutant tested for this position (i.e., C125D) appeared strongly reduced by the western-blot analysis."

Since C125 is located at the hydrophobic tetramerization interface, it is not surprising that the C125D mutation (introducing a negative charge) has a negative effect on function. I agree however that the positive effect of the C125A mutation is unexpected. A previous study has demonstrated that the relative expression level of P over the other proteins is crucial for optimal activity in a minigenome reporter assay (see figure 3A in: JOURNAL OF VIROLOGY, Nov. 2003, p. 11781–11789 DOI: 10.1128/JVI.77.21.11781–11789.2003). Is it possible that the increased activity of the C125A mutant is not directly related to the sequence alteration, but instead due to its lower expression level compared to the wild type?

4) line 304: "Altogether, these results suggest that the beta-layer-like motif is important for BoDV-1 replication, notably concerning D126 and H127 residues [...]"

Figure 2d gives me the impression that there is a salt bridge between the side chains of D126 and H127 from neighboring chains. Is this true? If so, this would explain that both these residues are functionally important because they help maintaining structural integrity of the tetramer. If indeed present, this salt bridge in my opinion deserves to be highlighted in figures 2d and/or 2e, as well as mentioned in the main text along with the mutagenesis results.

5) line 306: "Cysteine residues are known to form disulfide bridges and stabilise the structuration of proteins. Such a disulfide bridge has been previously proposed at position 125"
and line 465: "[...] we cannot exclude the potential presence of a disulfide bridges at position 125 [90] that would stabilise a dimer of BoDV-1, although it would not prevent the formation of the tetrameric form even if present."

Certainly, cysteines are known to form disulfides, but this mainly concerns proteins inside the lumen of the ER, the Golgi, or other organelles of the secretory pathway where the redox potential favors such bonds. Inside the cytoplasm or the nucleus, however, disulfides are probably a rare exception. Would the authors feel comfortable in making a respective remark in the context of the result from reference 90, which I personally believe is an experimental artifact due to oxygen exposure during sample processing? This may help putting things into perspective for those readers who are less familiar with biochemistry.

6) line 319: "The comparison with tetrameric oligomerisation domains of P from other Mononegavirales (metapneumovirus, parainfluenza 5, measles, mumps, Nipah and murine respirovirus) showed that BoDV-1 POD was indeed the longest."

In this context, I would appreciate a figure comparing the new BoDV-1 P structure to these formerly determined ortholog structures side-by-side.

7) line 521: "[...] non-nuclear interactors identified here would thus be irrelevant since P expressed alone would remain strictly nuclear and although biochemically able to form these proximal interactions, would never occur in the same cellular compartment upon actual BoDV-1 infection."

All bornaviral proteins, including P, are synthesised inside the cytoplasm (there are no ribosomes inside the nucleus!). Only after its synthesis, P is translocated to the nucleus. I therefore disagree that potential interaction partners from the cytoplasm must necessarily be irrelevant. Please reconsider.

8) Finally, I have a question regarding the BioID assay. As far as I understand from the supplementary methods, unspecific binding to BirA* has been accounted for by using a BirA* construct lacking the fused P moiety. Does this enzyme translocate to the nucleus, like P does? I encourage the authors to either argue based on their own experimental data, or to cite a respective reference, that their negative control is abundant in the nucleus of transfected cells and is therefore indeed an appropriate control for their experiment.
